# Generalized Resonance Sensor Based on Fiber Bragg Grating

Xinxin Chen [1,2], Enbo Wang [1,2], Yali Jiang [1,2], Hui Zhan [1,2], Hongwei Li [1,2], Guohui Lyu [1,2,*] and Shuli Sun [1,*]

1   College of Electronic Engineering, Heilongjiang University, Harbin 150080, China;
    chen.xinxin@hrbu.edu.cn (X.C.); 2201612@s.hlju.edu.cn (E.W.); 2171238@s.hlju.edu.cn (Y.J.);
    2000092@hlju.edu.cn (H.Z.); li_hw@hlju.edu.cn (H.L.)
2   National and Local Joint Engineering Research Center of Optical Fiber Sensing Technology,
    Heilongjiang University, Harbin 150080, China
*   Correspondence: lvguohui@hlju.edu.cn (G.L.); sunsl@hlju.edu.cn (S.S.)

**Abstract:** In response to the difficulty of weak detection of early bearing damage, resonance demodulation technology and the principle of fiber Bragg grating sensing strain were combined to design a fiber Bragg grating generalized resonance sensor, which can extract the weak pulse signal of weak detection of early bearing's early damage from rolling bearing. First, a principle of resonance dynamics of second-order mechanical systems based on fiber Bragg grating and generalized resonance principles is proposed. Second, the basic structure of the sensor is designed. Then, ANSYS finite element simulation is used to analyze the natural frequency of the sensor. Finally, the natural frequency value of the sensor was obtained through experiments. The experimental results of proof-of-principle show that the experimental results are consistent with the theoretical predictions. The theoretical model is accurate, which verifies the feasibility of the sensor.

**Keywords:** generalized resonance; fiber Bragg grating sensor; finite element simulation; free vibration





## 1. Introduction

At present, due to accidents caused by frequent bearing damage at home and abroad, early bearing damage monitoring and diagnosis is a current focus of research [1,2]. In the initial stage of bearing damage, damage vibration and shock are mainly distributed in the high-frequency range. This has a wide vibration frequency band, small amplitude and narrow width. The damage vibration information is relatively weak and easily covered by other vibration noise and environmental interference signals, which increases the difficulty of extracting damage information. Therefore, it is necessary to conduct effective identification and rapid diagnosis of early bearing damage signals [3,4].

The vibration analysis method is the most commonly used method in bearing damage monitoring and diagnosis methods, and its sensitivity is higher than that of temperature monitoring methods. However, the temperature monitoring method has a drawback: the bearing temperature changes significantly when the bearing damage is severe [5]. At present, the principle of bearing-damage vibration sensing is generally based on the principle of acceleration-sensing. In 2013, Yimin Shao of Chongqing University and others designed a three-dimensional mechanical differential piezoelectric acceleration sensor based on the piezoelectric principle, which can be used to obtain bearing damage vibration signals [6]. In 2017, Yi Zhang of Changzhi College proposed a MEMS acceleration sensor to realize multiple damage monitoring of motor deep groove ball bearings [7]. However, the electromagnetic acceleration sensor is not resistant to electromagnetic interference. As the working environment of the bearing is complex and changeable, the electromagnetic signal interferes with the vibration signal of the bearing to a certain extent and affects the diagnosis of bearing damage [8].

Fiber Bragg grating (FBG) has characteristics of anti-electromagnetic interference, high sensitivity, and high precision, so the optical fiber sensing technology also plays a huge role in the safety monitoring of bearing damage [9–12]. In 2018, Xiaofeng Wang of Shenzhen

University designed a synchronous monitoring sensor for the vibration temperature of train bearings based on FBG. The natural frequency of the sensor is 970 Hz [13]. This value is far less than the frequency value of the high-frequency component of the early bearing damage, and it cannot be used for early bearing damage monitoring and diagnosis.

Given that the natural frequency of the sensor is much lower than the frequency of the high-frequency component of the early bearing damage, the generalized resonance monitoring technology developed in recent years [14]. Generalized resonance monitoring technology for bearing damage is based on the generalized resonance phenomenon of an object after an instantaneous impact. When the instantaneous impact of bearing damage acts on the sensor, the instantaneous impact pulse contains a series of continuous frequency components from zero to infinity. The frequency component whose vibration frequency is equal to the natural frequency of the sensor affects the sensor. The sensor releases energy after absorbing energy [15]. Due to damping, the natural frequency of the sensor is free to dampen vibration. By analyzing its free attenuation vibration signal, its bearing damage can be diagnosed.

In 2012, Tang Deyao proposed a generalized piezoelectric resonance bearing damage monitoring sensor based on the principle of generalized resonance, which has a generalized resonance peak of 20 kHz and can monitor generalized resonance bearing damage. The sensor has good application prospects for the monitoring and diagnosing of early damage of vibrating bearing parts. The disadvantage is that the piezoelectric ceramic sensor based on the piezoelectric effect is not anti-electromagnetic. Bearing working environments are often very harsh, which greatly impacts the diagnosis result [16].

Based on the principle of generalized resonance combined with the principle of FBG sensing strain, an FBG generalized resonance sensor was designed. When the vibration component is damaged and causes an instantaneous impact, the sensor absorbs the impact energy and generates a generalized resonance wave. The component releases energy at the sensor's own high-frequency natural frequency. The FBG detects the vibration signal, and the high-speed FBG demodulator is used to obtain the center wavelength of the FBG. When the bearing is damaged early, a series of the periodic instantaneous impact energy is generated, and the time interval between the impact forces is extremely short, equivalent to the above-mentioned instantaneous impact. At the same time, the bearing working environment is harsh and is susceptible to electromagnetic interference. FBG has the advantage of anti-electromagnetic interference. Therefore, the sensor of this design can detect early bearing failure.

The main content of this article is as follows: This article first introduces the principle of resonance dynamics of second-order mechanical systems based on FBG and generalized resonance principles. Then the basic structure of the sensor is analyzed and designed. The Young's modulus of the bolt in the sensor structure was measured through experiments. Then, we used ANSYS finite element simulation software to analyze and obtain the natural frequency of the FBG generalized resonance sensor. Finally, the falling ball impact experiment was used to complete the measurement of the natural frequency of the sensor. The ANSYS simulation value of the natural frequency of the sensor designed in this design was close to the experimental measurement value. Both reached the high-frequency domain, and the resonance frequency was far from the low-frequency vibration frequency range. Moreover, the impulse response curve demodulated by FBG had no electromagnetic interference, and the spectrum was pure. Early fault detection of bearing signals is possible.

## 2. Principle

### 2.1. Principle of Generalized Resonance

Resonance means the phenomenon that the amplitude of a system increases significantly when the frequency of the excitation of the mechanical system is close to a certain order of the natural frequency of the system. At the resonance frequency, a small period of vibration can produce a large vibration because the system stores kinetic energy. When the resistance is extremely small, the resonance frequency is approximately equal to the

system's natural frequency or natural frequency. Generalized resonance describes the response of an object to an instantaneous assault. When the instantaneous assault acts on the object, the energy transfer of the impact force occurs, and the huge energy brought by the impact is stored. When the impact force disappears for an instant, the object releases energy with its own natural frequency, freely decayed vibration [17]. This process of energy storage and release is called generalized resonance, which can be regarded as a natural law of energy transfer and release. This has nothing to do with the shock frequency of the exciting force.

### 2.2. Principle of FBG Strain Sensing

When an incident light enters the FBG, the qualified light wave will continue to transmit through the FBG, and the unqualified part of the light will be reflected at the grating surface of the FBG [18,19]. The central wavelength $\lambda_B$ of the reflected wave of the FBG satisfies:

$$\lambda_B = 2n_{eff}\Lambda \tag{1}$$

In the formula: $\lambda_B$ is the central wavelength of the reflected light of the FBG, $\Lambda$ is the period of the grating region of the FBG, and $n_{eff}$ is the effective refractive index of the fiber core.

When the external conditions only change by stress or strain, the external force causes the FBG to expand and contract in the axial direction. The length and period of the FBG have changed. The relationship between length, period and strain are:

$$\Delta\varepsilon = \frac{\Delta L}{L} = \frac{\Delta\Lambda}{\Lambda} \tag{2}$$

In the formula: $\varepsilon$ is the axial strain of the FBG, $L$ is the effective length of the FBG.

At the same time, the FBG has an elastic-optical effect, and the change of the effective refractive index causes the change of the center wavelength of the reflected light. Under the influence of the elastic effect, the two sides of Equation (1) are differentiated and simplified to obtain:

$$\frac{\Delta\lambda_B}{\lambda_B} = (1 - p_e)\Delta\varepsilon \tag{3}$$

In the formula: $P_e$ is the effective elasticity coefficient of the FBG, which is related to the elasticity coefficient and Poisson's ratio.

Equation (3) is the theoretical formula for FBG strain sensing.

### 2.3. Monitoring Principle of FBG Generalized Resonance Sensor

When the vibrating component produces an instantaneous shock impacting the FBG generalized resonance sensor, the sensor absorbs energy and generates a generalized resonance response. At the same time, the FBG expands and contracts in the axial direction. Hence, the axial strain changes. It can be seen from Formula (3) that the change of the strain of the FBG can be converted into the change of the central wavelength of the FBG, which then reflects the change of external stress. By monitoring the dynamic change of the central wavelength of the FBG, the instantaneous impact of the vibrating component can be monitored.

The single-mass string longitudinal vibration model was used as the vibration sensor model, as shown in Figure 1 [20]. Assume that the vibrating mass $m$, displacement $x$, velocity $\dot{x}$, acceleration $\ddot{x}$, elastic force $kx$, damping force $c\dot{x}$, $c$ is the viscous damping coefficient, and $n$ is the attenuation coefficient.

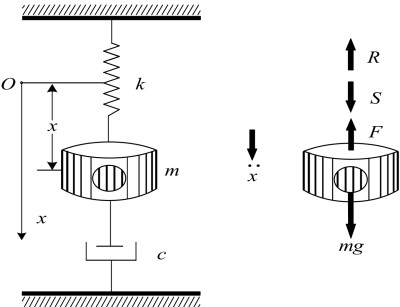

**Figure 1.** Mechanical second-order system model.

According to Newton's second law, a free vibration differential equation with viscous damping is established, and the simple resonance force acting on *m* is *Hsinωt*. The vertical downward direction of the x-axis is specified as the positive direction. Then, there is the following equation:

$$m\ddot{x}(t) + c\dot{x} + kx = H \sin \omega t \tag{4}$$

Substituting $\omega_n^2 = \frac{k}{m}$, $2n = \frac{c}{m}$, $h = \frac{H}{m}$ into Formula (4), we get:

$$\ddot{x} + 2n\dot{x} + \omega_n^2 x = h \sin \omega t \tag{5}$$

Its solution consists of two parts:

$$x(t) = x_1(t) + x_2(t) \tag{6}$$

The first part of the solution is the free attenuation vibration of the sensor subject to instantaneous excitation, which is only meaningful within a short period after the start of vibration, and will continue to attenuate as time increases; the latter part is forced vibration, which is called the stability of the system state solution.

The free decay equation of motion in the first part and its solution are:

$$\ddot{x} + 2n\dot{x} + \omega_n^2 x = 0 \tag{7}$$

$$x_1(t) = Ae^{-nt}\sin(\omega_d t + \alpha) \tag{8}$$

where $A = \sqrt{x_0^2 + \frac{(\dot{x}_0 + nx_0)^2}{\omega_d^2}}$, $tg\alpha = \frac{x_0\omega_d}{\dot{x}_0 + nx_0}$, $\omega_d = \sqrt{\omega_n^2 - n^2}$.

According to Formula (8), the corresponding amplitude free attenuation vibration waveform is obtained, as shown in Figure 2. $T_d$ is the free instantaneous vibration period.

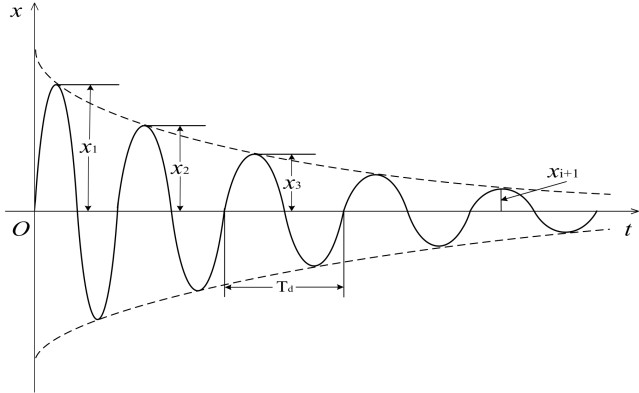

**Figure 2.** Freely attenuated vibration waveform.

The free damping vibration period $T_d$ is defined as:

$$T_d = \frac{2\pi}{\omega_d} = \frac{2\pi}{\omega_n} \frac{1}{\sqrt{1 - (\frac{n}{\omega_n})^2}} = \frac{T}{\sqrt{1 - \zeta^2}} \tag{9}$$

where $T = \frac{2\pi}{\omega_n}$ is the period of undamped free vibration, and $\zeta = \frac{n}{\omega_n}$ is the damping ratio, which is usually small and has little effect on the period.

When $t = t_i$ and $t = t_i + T_d$ are brought into Equation (8), take the natural logarithm of the ratio of the two amplitudes to obtain the attenuation coefficient $n$:

$$n = f_d ln \frac{x_i}{x_{i+1}}, f_d = \frac{1}{T_d} \tag{10}$$

The peak frequency of the amplitude-frequency characteristic curve of the free vibration signal of the sensor is the natural frequency of the sensor, which is used to test the simulation design of the sensor and verify the correctness of the designed resonance sensor.

When the free decay motion disappears, only forced vibration exists, so the motion equation of the sensor is:

$$x_2(t) = B \, sin(\omega t - \varphi) \tag{11}$$

In the formula: $\varphi$ is the phase difference, and $B$ is the vibration amplitude of the forced vibration of the sensor system.

Solving Equation (9) based on the complex exponent, we get:

$$x_2(t) = \overline{B} e^{j\omega t} \tag{12}$$

where:

$$\begin{aligned} \overline{B} &= \frac{h}{\omega_n^2 - \omega^2 + j2n\omega} \\ B &= \frac{a}{\sqrt{(\omega_n^2 - \omega^2)^2 + (2n\omega)^2}} \end{aligned} \tag{13}$$

$B$ is the modulus of $\overline{B}$ amplitude, with:

$$B = \frac{h}{\sqrt{(\omega_n^2 - \omega^2)^2 + (2n\omega)^2}} \tag{14}$$

Equation (14) is the calculation expression of the steady-state amplitude $B$ of the spring–mass model of the fiber grating vibration sensor.

When the external conditions only change by stress or strain, the external force causes the FBG to expand and contract in the axial direction. The axial expansion and contraction change is $\Delta\varepsilon$. According to Formula (2), the displacement change of the single-particle string longitudinal vibration model under the action of external force is $\Delta L$:

$$\Delta\varepsilon = \frac{x}{L} \tag{15}$$

Substituting Equation (15) into Equation (3). The relationship between the center wavelength of the FBG and the displacement of the vibration model can be obtained:

$$\frac{\Delta\lambda_B}{\lambda_B} = (1 - p_e) \frac{x}{L} \tag{16}$$

## 3. Design and Material Selection of FBG Generalized Resonance Sensor

*3.1. Sensor Package Design*

As shown in Figure 3, the entire FBG resonant sensor was mainly composed of protective housing, an optical fiber port, a resonator, and a base. The resonant body comprises a mass block, an elastomer and a bolt. The mass block was located on the top

and provided gravity loading for the sensor. The elastomer was located below to provide restoring force for the mass to vibrate back to the equilibrium position. The bolt penetrated the mass block and the elastomer body vertically. There was a vertical hole inside the bolt, which was mainly used for curing FBG [20]. The FBG was cured into a whole through the resin and the mounting bolt, which was used to sense the resonance frequency of the second-order mechanical system and the resonance amplitude of the mass block.

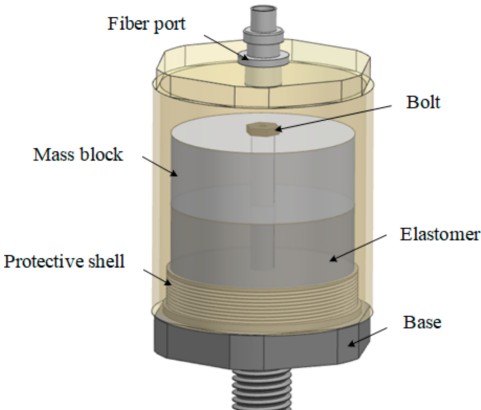

**Figure 3.** Basic sensor structure.

The structure of the sensor resonator is shown in Figure 4. Among them, the material of the mass block was determined to be 316L stainless steel with high-density and high quality; the elastomer had the effect of improving the sensitivity of the sensor. The material was made of polyurethane with high resilience and corrosion resistance.

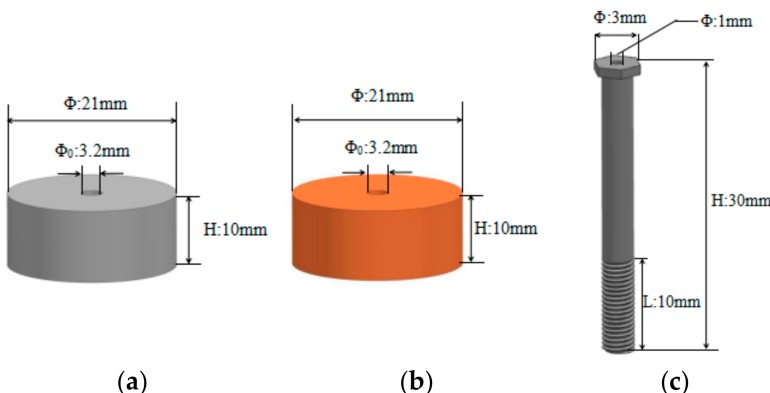

**Figure 4.** Resonance body structural parts, (**a**) mass block (**b**) elastomer (**c**) bolt.

The specific parameters were as follows:

Mass block and elastomer: H = 10 mm, $\Phi$ = 21 mm, $\Phi_0$ = 3.2 mm; bolts: H = 30 mm, $\Phi$ = 3 mm, $\Phi_0$ = 1 mm; threads under the bolts: R = 1 mm, $\alpha$ = 2°, L = 10 mm (wherein H stands for height, $\Phi$ stands for diameter, $\Phi_0$ stands for vertical center hole diameter, R stands for pitch, $\alpha$ stands for thread angle, and L stands for thread length).

Considering the harsh working environment of the sensor, to ensure that the FBG has good sensing performance in its environment, the FBG in the sensor needs to be packaged and protected. Inside the sensor, a two-component acrylic adhesive was used to fix the FBG on the inner wall of the bolt hole to ensure that it could sensitively sense the vibration damage signal. Outside the sensor, a nylon sleeve was placed on the bare fiber, which could effectively reduce the influence of external forces on the measurement process.

### 3.2. Determination of Young's Modulus of Bolt Metal Material

In the sensor structure of this design, the bolt is used to solidify and encapsulate the FBG sensitive element. Selecting the Young's modulus value of the material is extremely important. The Young's modulus value of an object can reflect the stiffness of the material itself. It has a great influence on the natural frequency of the sensor designed in the article. In the modal simulation analysis of the sensor, it is also necessary to accurately substitute Young's modulus value of the bolt material.

For this, the following measurement methods were used to measure Young's modulus of the material. The metal material was fixed on the workbench, and vertical upward pressure *F* was applied to one end of the metal material. The other end of the metal material was provided with a pressure sensor to measure the pressure of the metal material. The FBG was solidified on the surface of the metal material, and the strain generated by the metal material was converted into the axial strain of the FBG. When there was pressure, the FBG on the surface of the metal material changes the center wavelength due to deformation. We used an FBG demodulator to monitor the change in the center wavelength of the reflection spectrum of the FBG and obtain the difference between the center wavelength change and the strain value of the FBG. According to the relationship between the force *F* and Young's modulus and strain $F = E \cdot S \cdot \varepsilon$, Young's modulus of the material was measured.

Through the above measurement method, Young's modulus of the bolt metal material was $203.466 \times 10^9$ N/m$^2$. This value was used in the modal simulation to calculate the natural frequency of the sensor.

### 4. Experiment and Analysis Results

To verify the correctness of the designed sensor, it is necessary to perform natural frequency simulation measurements and analyses on the sensor. Because the natural frequency is an inherent property of matter, it only depends on the nature of the material itself and will not change due to external influences. When the system is subject to an instantaneous impact, it will vibrate at its natural frequency. This design adopts the free vibration method to realize the measurement of natural frequency.

### 4.1. Natural Frequency Simulation Analysis of FBG Generalized Resonance Sensor

ANSYS finite element simulation software was used to analyze the natural frequency of the FBG generalized resonance sensor. To simplify the process of extracting the natural frequency of the sensor, the simulation used the resonant body structure of the sensor for simulation. First, we established a three-dimensional model of the resonance body structure. The resonant body parameters were the same as Section 3.1.

The modeled resonant body structure was imported into ANSYS for modal simulation analysis, and the natural frequencies of the 1–6 order vibration models were obtained, which were 885 Hz, 895 Hz, 1367 Hz, 6830 Hz, 6857 Hz, and 9770 Hz, respectively, as shown in Figure 5.

According to the size of the three-dimensional modeling mass, the mass of the mass block was calculated to be 26.9 g, and the material was stainless steel, the stiffness value $K = 4.75 \times 10^7$ N/m. The natural frequency calculation formula is:

$$f = \frac{1}{2\pi}\sqrt{\frac{K}{m}} \tag{17}$$

Theoretical calculations showed that the natural frequency of the resonator structure was 6691 Hz. The theoretical calculation value was close to the simulated fourth-order natural frequency of 6830 Hz.

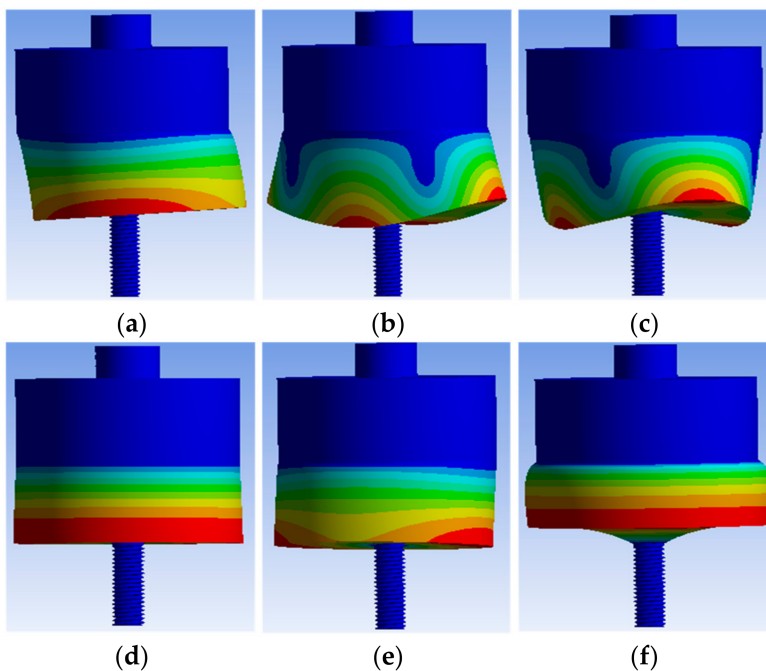

**Figure 5.** Vibration models, (**a**) one modal (**b**) two modal (**c**) three modal (**d**) four modal (**e**) five modal (**f**) six modal.

*4.2. Design of Natural Frequency Experiment Device for FBG Generalized Resonance Sensor*

To measure the natural frequency of the sensor, the free vibration method was used. As shown in Figure 6, the measuring device for measuring the natural frequency by the free vibration method comprised a steel plate, steel ball, FBG generalized resonance sensor, high-speed FBG demodulator, and signal acquisition software. The impact source used a steel ball drop. The steel ball fell freely on the steel plate to produce an instantaneous impact. The shock wave was transmitted through the steel plate to the FBG generalized resonance sensor. The instantaneous impact energy of the steel ball was transmitted to the sensor, and the sensor freely dampened the vibration at its natural frequency. In the whole process, the curing and encapsulating of the FBG in the resonant body was recorded by the free attenuation vibration signal of the sensor.

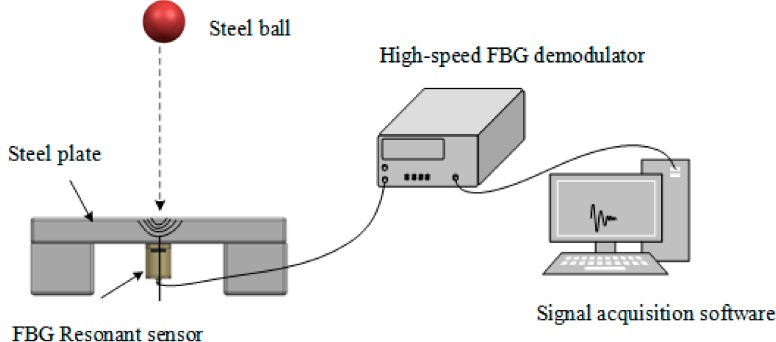

**Figure 6.** Natural frequency measuring device's design picture of free vibration method.

*4.3. Analysis of Natural Frequency Test Data*

The experimental test adopted the falling ball impact method. Different experimental data were obtained by using the method of control variables.

### 4.3.1. Control the Diameter of the Steel Ball Unchanged

The diameter of the steel ball was controlled to 20 mm so that the steel ball fell from different heights to impact the sensor.

A steel ball with a diameter of 20 mm was used to impact the steel plate. The steel ball freely fell to produce an instantaneous impact. The shock wave was transmitted through the steel plate to the FBG generalized resonance sensor. In other words, the short-term impact energy of the steel ball was transmitted to the sensor. The sensor attenuated the vibration freely with its natural frequency; curing the FBG encapsulated in the resonant body records the free attenuation vibration signal of the sensor. In other words, the center wavelength of the FBG changed, and then used the high-speed FBG demodulator to obtain the center wavelength of the FBG. Amplitude-frequency analysis and power spectrum analysis were performed on the central wavelength of the FBG, and finally, the natural frequency of the sensor was obtained.

The time-domain characteristic curve of the steel ball falling from the height of 10 cm and 30 cm to the steel plate is shown in Figure 7. From Figure 7, it can be seen that when the steel ball fell from different heights, the sensor obtained different impact energies. Both underwent free damping vibration, and the pulse width was 6 ms, the sensor generated a generalized resonance wave.

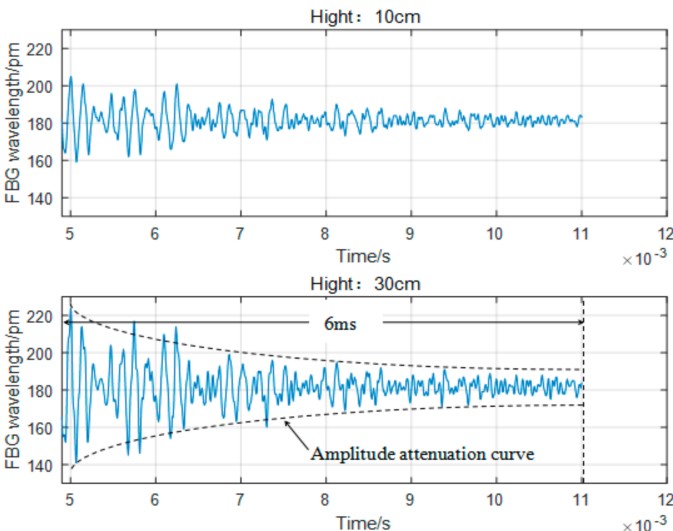

**Figure 7.** 10 cm and 30 cm time-domain characteristic curves.

The amplitude-frequency analysis of the experimental data is shown in Figure 8. It can be seen in the figure that the maximum frequency is the measured natural frequency of the sensor. When the drop height of the steel ball was 10 cm, the measured natural frequency value of the sensor was 6362 Hz. When the drop height of the steel ball was 30 cm, the measured natural frequency value of the sensor was 6271 Hz. When steel balls with different drop heights impacted the sensor, the natural frequency values of the sensor were the same. It was close to the ANSYS simulation fourth-order natural frequency of 6830 Hz.

Perform power spectrum analysis on the experimental data, as shown in Figure 9. It can be seen from the figure that when the drop height was 10 cm, the maximum energy corresponding to the natural frequency 6362 Hz was 4796 dB. When the drop height was 30 cm, the maximum energy corresponding to the natural frequency 6271 Hz was more than 10,000 dB. We could see that the impact energy absorbed by the FBG generalized resonance sensor was different for different falling heights. Under different impact energy, generalized resonance with the vibration frequency of its natural frequency was produced.

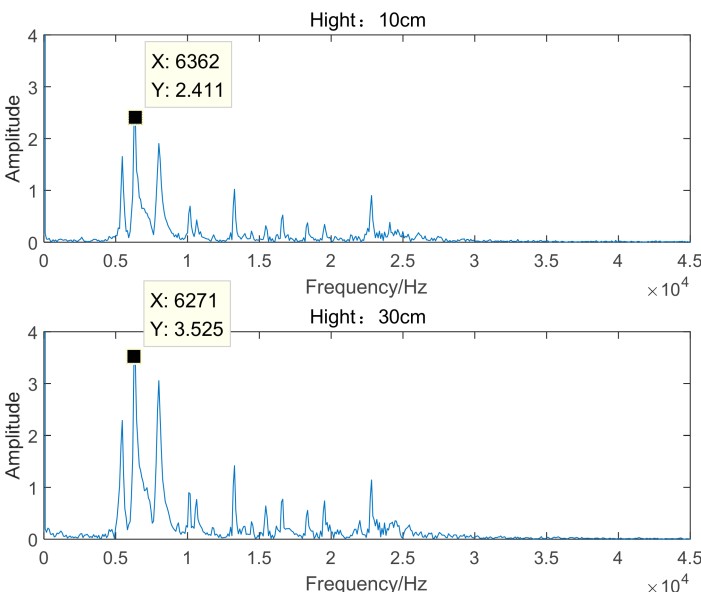

**Figure 8.** 10 cm and 30 cm amplitude-frequency characteristic curves.

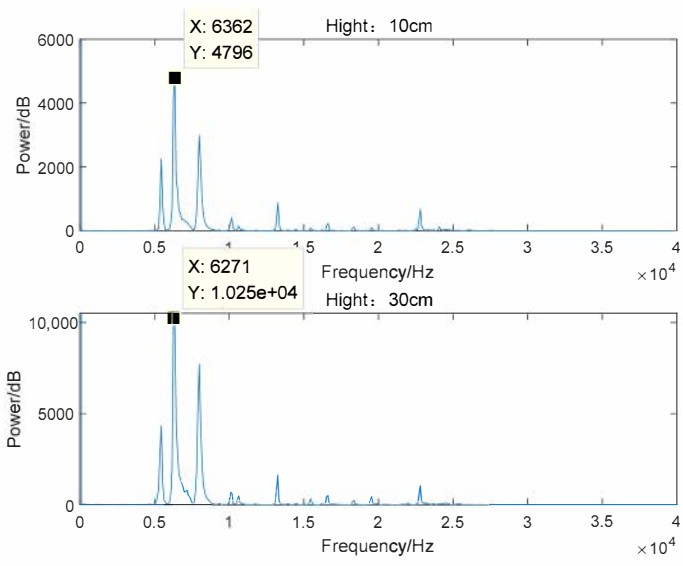

**Figure 9.** Impact power spectrum curves of 10 cm and 30 cm drops.

4.3.2. Control the Drop Height of the Steel Ball Unchanged

The drop height of the steel ball was controlled to 20 cm. The time-domain characteristic curve of steel balls with diameters of 8 mm, 10 mm and 20 mm is shown in Figure 10, which shows that when the quality of the steel ball was different, the sensor obtains different impact energy. When we carried out free attenuation vibration, the pulse width was 6 ms, the sensor produced a generalized resonance wave.

The amplitude-frequency analysis of the experimental data is shown in Figure 11. From the figure, it can be seen that under the impact of steel ball diameters of 8 mm, 10 mm, and 20 mm, the natural frequency values of the obtained sensors were 6271 Hz, 6362 Hz and 6362 Hz, respectively, which were close to the ANSYS simulation fourth-order natural frequency of 6830 Hz.

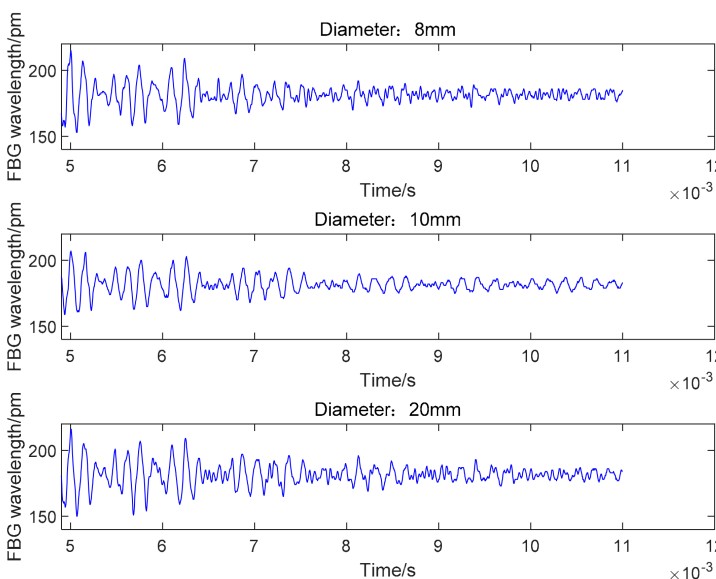

**Figure 10.** Time-domain impact characteristic curves of steel balls with different diameters.

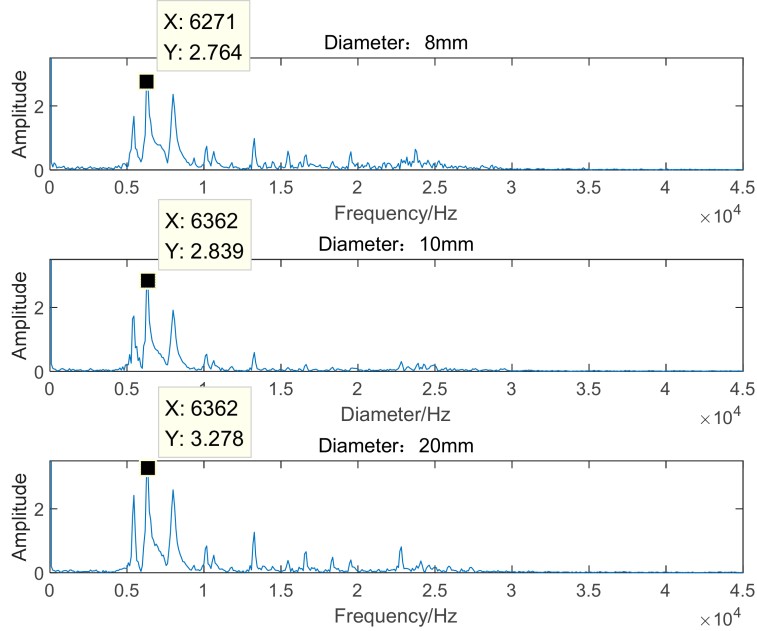

**Figure 11.** Impact amplitude-frequency characteristic curves of steel balls with different diameters.

Perform power spectrum analysis on the experimental data, as shown in Figure 12. It can be seen in the figure that under the impact of steel balls with diameters of 8 mm, 10 mm and 20 mm, the corresponding maximum energy values at natural frequencies were 6303 dB, 6654 dB and 8869 dB. It was verified that when steel balls of different quality impact the sensor, the impact energy absorbed by the sensor was also different. However, the natural frequency test values were similar. Both were close to the ANSYS simulation fourth-order natural frequency of 6830 Hz. This proved that the generation of generalized resonance had nothing to do with the impact energy, and the natural frequency of the sensor was its nature and had nothing to do with the impact energy.

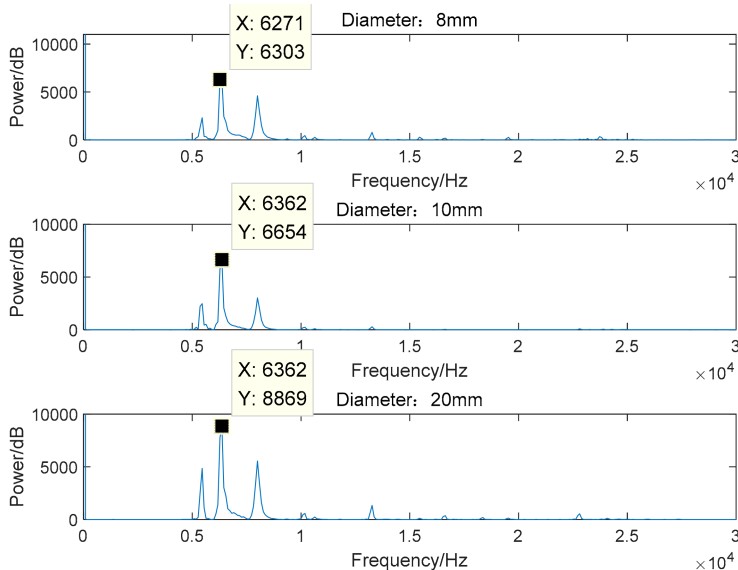

**Figure 12.** The power spectrum curves of steel balls with different diameters.

According to the analysis of the experimental data of the falling ball instantaneous impact sensor, it was observed that when the steel ball hits the FBG generalized resonance sensor with different masses and different falling heights, free damped vibration was generated. The pulse width was 6 ms. The vibration waveform was a generalized resonance wave. The frequency spectrum of its amplitude-frequency characteristic curve was pure, and there was no electromagnetic interference phenomenon. The natural frequency values of the sensor analyzed by the amplitude-frequency characteristic curves were 6362 Hz and 6271 Hz, respectively. Close to the theoretical value of 6691 Hz. It was also close to the fourth-order natural frequency value of 6830 Hz simulated by ANSYS. According to the analysis of the power spectrum characteristic curve, the natural frequency of the sensor was its character and had nothing to do with the impact energy. The generation of generalized resonance was also independent of impact energy. This paper mainly studies the early damage monitoring method of bearing based on FBG. The principle was verified by the instantaneous impact experiment of steel balls because the steel ball instantaneous impact sensor was equivalent to the instantaneous impact of the early damage of the bearing to the sensor. The experimental results show that the experimental data are consistent with the theoretical predictions, and the theoretical model was accurate. It could be used to extract high-frequency impulse response signals caused by bearing damage and could be anti-electromagnetic interference.

## 5. Discussion

The research in this paper was mainly aimed at detecting early damage signals of bearings. Vibration and shock caused by early bearing damage have high-frequency and instantaneous characteristics. As the damage intensified, the frequency of damage vibration and shock gradually shifted to low frequencies. In addition, the working environment of the bearing is harsh, there will be interference signals, and other vibrations and noise can easily conceal the early bearing damage signals. This study combines the FBG principle and the generalized resonance principle to design a sensor that will cause resonance after a small impact. The resonance frequency reached more than 6000 Hz, and the resonance frequency was far from the low-frequency vibration frequency range, which could detect high-frequency signals well. Avoid the interference of low-frequency vibration. Moreover, the natural frequency was its inherent characteristic and had nothing to do with the impact energy. The sensor studied in this paper could perform online sensing measurement of high-frequency signals of bearing damage and impact response. As a sensing detection

unit, FBG had the advantages of anti-electromagnetic interference, the measurement signal was pure, and it was convenient for data analysis and processing.

## 6. Conclusions

An FBG generalized resonance sensor was designed based on the principle of resonance demodulation and the principle of FBG sensing strain. As a sensing unit, FBG had good anti-electromagnetic interference ability. When the vibrating part of the sensor was subjected to an instantaneous impact, the sensor absorbed the impact energy to generate a generalized resonance wave and releases energy at its natural frequency to dampen vibration freely. The natural frequency of the sensor of this design reached the high-frequency range, which could detect high-frequency vibration signals. The experimental results of proof-of-principle show that the natural frequency of the sensor was its inherent characteristic and had nothing to do with the impact energy. Different instantaneous impact energy caused the sensor to produce generalized resonance, and the generalized resonance wave had nothing to do with the impact energy. The sensor can extract the weak pulse signal of the early failure of the vibrating component, which lays the foundation for the early diagnosis of the damage of the bearing component in the mechanical equipment.

**Author Contributions:** Conceptualization, S.S. and G.L.; software, H.Z.; validation, Y.J.; formal analysis, H.L.; writing—original draft preparation, E.W.; writing—review and editing, X.C.; All authors have read and agreed to the published version of the manuscript.

**Funding:** This research was funded by Heilongjiang Provincial Science and Technology Department, grant number GZ11A306, Heilongjiang University, grant number HDJMRH201904.

**Conflicts of Interest:** The authors declare no conflict of interest.

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
