# Peer review of "Generalized Resonance Sensor Based on Fiber Bragg Grating"

_photonics, doi:10.3390/photonics8050156_

Round 1

Reviewer 1 Report

The article presented the theory, design and analysis of generalized resonance sensor using FBG. It is unclear about the novelty of the article. From line 55 - 119, thorough theory of generalized resonance is presented. However, it is unclear whether it is just general theory or any expansion of the theory by the authors. 

Experimental verification of the design will make the paper stronger. 

Author Response

First of all, I would like to thank the reviewers for their efforts and suggestions for revision. 

This theory is a typical mechanical second-order system theory, (corresponding to references [[17]-[20]). The fiber Bragg grating is cured into a whole through the resin and the mounting screw, which is used to sense the resonance frequency of the mechanical second-order system and the resonance amplitude of the mass block. It can carry out online sensing measurement of high frequency signals of bearing damage and impact response. The test method and sensor structure have not yet seen relevant reports.

In this revised version, we have implemented all the suggestions from reviewers, and made detailed changes. The main changes can be seen as gray text in the revised manuscript.

Reviewer 2 Report

The article presents theoretical modeling and experimental results of observing the resonant frequency of the vibration sensor model. In my opinion this work looks like the student laboratory workshop, but not a serious scientific work. Unfortunately, I was unable to find any new interesting results what could be published in Photonics journal. My conclusion is based on the following reasons:

  1. The second part of the article deals with the basic concepts from a physics textbook, such as resonance, the basic equations of an oscillatory system, and the principle of operation of the Bragg grating
  2. The third part describes the design of the vibration sensor. Indeed, the presented design can be the basis for a vibration sensor with a natural vibration frequency. However, why exactly such a construction and with such given parameters? But we can assume that this design was chosen for convenience. This is acceptable.
  3. The next part provides a simulation of the behavior of a specific sensor design and further experimental work to confirm the presented model. What are the conclusions from these results? What is the scientific value of the data presented? I understand when a specific engineering structure is modeled, load tests and vibration resistance are performed. The theoretical calculations of complex engineering structures, the practical value of which is to create reliable and predictable elements, are confirmed. It would also be understandable if the paper proposed a new model for calculating critical parameters, applicable to various types of structures. But what is new in the results and what is their practical value?
  4. And finally, what does the Bragg gratings have to do with it?

In my opinion the paper cannot be published in the Photonics journal.

Author Response

First of all, I would like to thank the reviewers for their efforts and suggestions for revision.

  1. Since the structure of the sensor is a typical mechanical second-order system, it is necessary to introduce the principle of resonance dynamics of the mechanical second-order system. At the same time, the fiber Bragg gratingis cured into a whole through the resin and the mounting screw, which is used to sense the resonance frequency of the mechanical second-order system and the resonance amplitude of the mass block. It can carry out online sensing measurement of high frequency signals of bearing damage and impact response. Therefore, it is necessary to introduce the working principle of fiber Bragg grating.
  2. This paper proposes a sensing method for bearing damage and impact response measurement. A theoretical model of sensor structure design for sensing the resonance signal of the mechanical second-order system with fiber Bragg grating is established. The sensor structure and parameters are designed based on theoretical models. In order to avoid the interference of low frequency vibration, the design resonance frequency is far away from the low frequency vibration frequency range. Based on this, the mass m and the elastic body stiffness coefficient k are determined.
  3. (1) The experimental results of the principle verification show that the experimental results are consistent with the theoretical predictions, and the theoretical model is accurate, which can be used to extract the shock response signal caused by damage.

 (2) The experimental methods and test methods are feasible. The impulse response curve demodulated by fiber Bragg grating has no electromagnetic interference and the frequency spectrum is pure.

 (3) Therefore, the mechanical second-order system resonance sensor based on fiber Bragg grating can be applied in the field of bearing monitoring and has certain application value.

  1. The fiber Bragggrating in the sensor is cured into a whole through the resin and the mounting screw, which is used to sense the resonance frequency of the mechanical second-order system and the resonance amplitude of the mass block. It can carry out online sensing measurement of high frequency signals of bearing damage and impact response.

In this revised version, we have implemented all the suggestions from reviewers, and made detailed changes. The main changes can be seen as gray text in the revised manuscript.

Reviewer 3 Report

In response to the actual needs of early detection of bearing damage, the paper proposes a new method based on fiber Bragg grating to detect generalized resonance signals, which replaces the traditional piezoelectric measurement sensor, and has certain innovations in the measurement method and sensor structure design. Theoretical analysis, numerical simulation, and experimental results all prove the feasibility of the scheme, which is expected to solve the anti-electromagnetic interference problem of the piezoelectric generalized resonance sensor and has certain application prospects.

The paper has the following problems that need to be corrected.

  1. The mixed-use of fiber grating, fiber Bragg grating, and FBG in many places in the paper is confusing and needs to be modified to FBG throughout the article.
  2. The structure of the sensor in Figure. 3 needs to be marked with geometric dimensions.
  3. 2. Determination of Young's modulus of bolt metal material. This part of the description of Young's modulus measurement is not clear, and the corresponding schematic description needs to be added.
  4. The short-term impact energy used in the article is not appropriate enough to describe transients and needs to be modified.
  5. The expressions of anti-electromagnetic interference in lines 38 and 41 should be unified.
  6. The font size in the figure needs to be uniform.
  7. The font size of the formula needs to be adjusted and formatted.
  8. English writing needs improvement.
  9. The paper needs to add that the impact of the steel ball can be equivalent to the impact caused by the early damage of the bearing, and the generalized resonance excited by the damaging impact can be used for bearing damage monitoring.

It is suggested that this paper can be published after minor revision.

Author Response

First of all, I would like to thank the reviewers for their efforts and suggestions for revision.

  1. We have unified the abbreviation of fiber Bragg grating. We appreciate your valuable suggestions.   
  2. We have marked with geometric dimensions, which can be seen in revised manuscript. Thank you for your valuable suggestions.
  3. The main subject of this work is to report the novel bearing damage measurement method for the first time, specially, not just for the measurement of Young's modulus. Based on the theoretical model, we can select the appropriate material.
  4. We have accepted your suggestion.
  5. We have accepted your suggestion.
  6. We have properly regulated the text size in all the figures, which can be seen in revised manuscript.
  7. We have properly regulated the text size in all the formula, which can be seen in revised manuscript.
  8. We have revised some spelling and grammar errors. We appreciate your valuable suggestions.
  9. This paper mainly studies the early damage monitoring method of bearing based on fiber Bragg grating, and carries out the principle verification experiment. The results of the steel ball impact simulation experiment are consistent with the theoretical prediction, and the theoretical model is accurate. The sensor studied in this paper can be used to extract the shock response signal caused by damage. Moreover, fiber Bragg grating sensing has the advantages of anti-electromagnetic interference, and the measurement signal is pure, which is convenient for data analysis and processing.

In this revised version, we have implemented all the suggestions from reviewers, and made detailed changes. The main changes can be seen as gray text in the revised manuscript.

Round 2

Reviewer 1 Report

The clarity of the revised article has much improved in comparison with the first draft. A few recommendations: 

1) the introduction section can be more concise with an emphasis on the novelty of the present work. 

2) the format of the article can be improved such as line 361-379. 

Author Response

First of all, I would like to thank the reviewer for the suggestions for revision. The main changes can be seen as red underlines in the revised manuscript.

  • The introduction section has been revised, and the content is concise, focusing on the novelty of this work.
  • The format of the article has been improved, such as lines 361-379.

Reviewer 2 Report

The introduction is written in detail. However, my opinion about the content of the work remains the same. I have not found any fundamentally new approaches here. Or the authors couldn't show it. However, the proposed approach can be used and, if the editor and other reviewers consider this article worthy of publication, I can agree with them. However, to further improve the introduction, I recommend that the authors read the following papers (for example):

10.1002/stc.1492

10.3390/s18103395

10.1016/j.engstruct.2005.09.018

10.1016/j.ymssp.2018.10.035

10.1177/1475921710365437

Author Response

 First of all, I would like to thank the reviewer for the suggestions for revision.

Thank you very much for the reference articles provided by the experts, but these references mostly use fiber Bragg gratings for mechanical vibration and ultrasonic sensing applications. It is essentially different from this article. In this paper, the shock excitation energy caused by the early damage of the bearing is absorbed by the mechanical second-order system, and the absorption energy is released by the vibration of the natural frequency of the system. The fiber Bragg grating is used to sense the resonant frequency of the mechanical second-order system. This is also the new development of generalized resonance sensing by replacing piezoelectric ceramic sensors with fiber Bragg gratings in this article. It can realize the measurement of high-frequency vibration of bearing damage and impact response, avoids the interference of low-frequency vibration, effectively improves the anti-interference ability of generalized resonance sensing, and is more suitable for engineering applications.